

# Effect of insecticide formulation and adjuvant combination on agricultural spray drift

Collin J. Preftakes[1], Jerome J. Schleier III[2], Greg R. Kruger[3], David K. Weaver[1] and Robert K.D. Peterson[1]

[1] Land Resources and Environmental Sciences, Montana State University, Bozeman, MT, USA
[2] Corteva[TM] Agriscience, Indianapolis, IN, USA
[3] West Central Research & Extension Center, University of Nebraska, North Platte, NE, USA

## ABSTRACT

Loss of crop protection products when agricultural spray applications drift has economic and ecological consequences. Modification of the spray solution through tank additives and product formulation is an important drift reduction strategy that could mitigate these effects, but has been studied less than most other strategies. Therefore, an experimental field study was conducted to evaluate spray drift resulting from agricultural ground applications of an insecticide formulated as a suspension concentrate (SC) and as a wettable powder (WP), with and without two adjuvants. Droplet sizes were also measured in a wind tunnel to determine if indirect methods could be substituted for field experimentation to quantify spray drift from these technologies. Results suggest that spray drift was reduced by 37% when comparing the SC to the WP formulation. As much as 63% drift reduction was achieved by incorporating certain spray adjuvants, but this depended on the formulation/adjuvant combination. The wind tunnel data for droplet spectra showed strong agreement with field deposition trends, suggesting that droplet statistics could be used to estimate drift reduction of spray solutions. These findings can be used to develop a classification scheme for formulated products and tank additives based on their potential for reducing spray drift.

## INTRODUCTION

Spray drift from agricultural applications of pesticides is an expected outcome, regardless of measures to minimize its occurrence (*Damalas, 2015*; *EPA, 1999*; *Felsot et al., 2011*; *Salyani & Cromwell, 1992*). The U.S. Environmental Protection Agency (EPA) defines drift as the "movement of pesticide dust or droplets through the air at the time of application or soon after, to any site other than the area intended" (*EPA, 2016b*). Loss of crop protection products via drift can result in potentially harmful human and environmental health effects, inefficient pest control, and economic losses to the product user. Pesticide drift is axiomatically problematic because it compromises the objectives of integrated pest management, which are to reduce pest status through means that are effective, economically sound, and ecologically compatible (*Pedigo, 1989*). Developments in drift

Corresponding author
Collin J. Preftakes, collin.preftakes@student.montana.edu

reduction technologies (DRTs) and environmental policy for pesticides have progressed through increased knowledge of drift phenomena, but important research gaps remain.

Environmental conditions can affect pesticide spray drift and must be considered when making an application, but these cannot be controlled. The only option when facing unfavorable environmental conditions is to decide to postpone or cancel the application. Operating conditions, on the other hand, can be manipulated to mitigate spray drift by the person making the application decision. Environmental conditions such as wind speed and direction, temperature, relative humidity (RH), atmospheric stability, and crop characteristics interact with airborne droplets and influence their deposition. Operational parameters such as boom height, driving speed, spray pressure, nozzle orientation, and application rate can reduce drift and are among the variables that can be readily manipulated by the equipment operator. Nozzle type, nozzle size, formulation type, and tank additives are commercially available DRTs designed to decrease drift through modification of the droplet size distribution upon atomization. Other DRTs such as shielded- and air-assisted sprayers function by interrupting the interaction between airborne droplets and the surrounding air movement.

There are a number of commercially available DRTs and the EPA has recently developed a protocol for verifying and rating their drift reduction potential (*EPA, 2016a*). The protocol provides a standard method for the application technology industry to voluntarily test DRTs. Pesticide drift considerations are included in all registration processes and registrants are encouraged by EPA to include verified DRT options on product labels. However, there are limitations to this protocol because it has only been evaluated for spray nozzles in low- and high-speed wind tunnels, and does not include the effect of tank mixes.

In this paper, the combination of a formulated pesticide active ingredient, with or without an adjuvant, is referred to as the spray solution. A pesticide formulation is a mixture of chemicals designed to maximize intended biological efficacy. Physical properties of certain formulation types have been shown to influence how droplets are formed upon atomization at the spray nozzle. Adjuvants are tank additives that are marketed for their enhancement benefits according to the function they are designed to perform. Some adjuvants are designed to enhance the performance of the pesticide, usually through better absorption, whereas others are designed to enhance qualities of the spray by modifying the physical properties of the spray solution (*De Oliveira et al., 2013*; *Richards, Gripp & Riden, 2017*).

Manipulating components in the spray solution as a drift reduction strategy has been reported in the scientific literature less than other technologies such as nozzle type, and results are variable (*Butler Ellis & Bradley, 2002*; *De Schampheleire et al., 2009*; *Miller, Butler Ellis & Lane, 2011*; *Stainier et al., 2006*; *Al Heidary et al., 2014*). Quantification of drift reduction due to formulation and adjuvant type is an important objective because these components can have effects that are equivalent to those due to nozzle type on downwind deposition (*Miller, Butler Ellis & Lane, 2011*). Furthermore, selecting the formulation with optimal drift reduction potential could reduce the need to include drift reduction adjuvants, which may not perform consistently when employed in different

combinations. Therefore, this research characterizes downwind deposition of two common formulation types and adjuvants from a ground sprayer in a three-year field study.

## MATERIALS AND METHODS

### Field trial design

Field experiments were conducted over three consecutive summers from 2014 to 2016 at the Dow AgroSciences Western U.S. Research Center near Fresno, California. The topography at the field site was flat, and there was no vegetation because it had been fallow the previous season, and disked before the start of the experiment. The spray swath was 145-m long by 15-m wide, and oriented with driving direction perpendicular to the wind direction. The off-target area was 110 by 145 m and downwind from the spray source. Two sample lines 2.5 m to the left and right of the center of the off-target plot consisted of both horizontal (ground) and vertical (one and two m above ground) sample locations. The sample lines were perpendicular to the spray line and approximately parallel to the wind direction. The orientation and relative lengths of the spray swath and sample lines (Fig. 1) were designed so that spray droplets could travel toward the farthest downwind sample locations if no more than a 30° deviation angle in wind direction was allowed (ASABE, 2009a).

Downwind insecticide ground deposition was collected with 14-cm diameter plastic Petri dishes (Cat. No. FB0875714; Fisher Scientific, Hampton, NH, USA). Petri dishes were horizontally placed on plywood at 8 distances of 1, 2, 4, 8, 16, 32, 64, and 110 m from the field edge, along each of the two sample lines. The downwind samplers along the two lines were subsamples, so for statistical analysis the deposition data were averaged for each distance. Plywood was placed on the ground to provide a level surface for the dishes. Also, along the two sample lines (averaged for statistical analysis) at 2, 8, 32, 64, and 110 m, vertical samplers were positioned one and two m above the ground to sample the size distribution of the spray droplets at different heights and distances. Each vertical sampler (spinner) consisted of two rotating microscope slides with a spin rate of 600 rpm, designed to impinge airborne droplets (Leading Edge Associates, Fletcher, NC, USA). The microscope slides were coated with a magnesium oxide (MgO) powder so that analysis could be done at a later date. This is made possible by the MgO–coating because measurements are made on the droplet imprints instead of the droplets themselves, which are prone to evaporation (Chaskopoulou et al., 2013).

An untreated area located 15-m upwind from the spray swath contained negative controls for both horizontal and vertical samplers, and a weather station to monitor environmental conditions. The weather station consisted of a Hobo Micro Station Data Logger (Onset Computer Corporation, Bourne, MA, USA) attached to 12-bit temperature and RH sensors with a solar radiation shield and a wind speed and direction smart sensor positioned 2.5 m above the ground. The Hobo wind speed smart sensor (S-WSA-M003) had a starting threshold of ≤1 m/s and the logger was set to measure wind speed every 20 s. The average wind speed was calculated over the first 10 min of each treatment replication to characterize wind speeds before, during, and after each application.

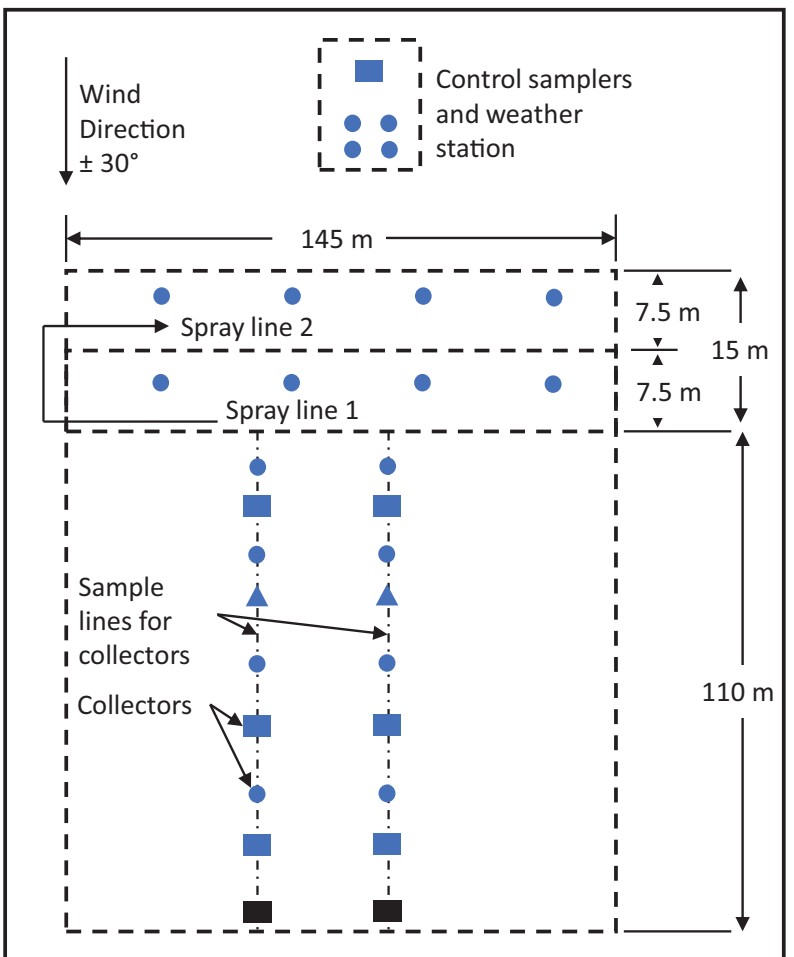

**Figure 1 Field layout for drift experiment in Fresno, California.** Blue circles represent locations where only horizontal ground samplers (14-cm diameter Petri dishes) were placed. Blue triangles represent locations where ground samplers and two-m vertical samplers (rotating impingers) were placed. Blue rectangles represent locations were ground samplers and one- and two-m vertical samplers were placed. Black rectangles (located at farthest downwind distance from spray source) represent locations were only one- and two-m vertical samplers were placed, no ground samplers were placed there in 2015 and 2016. The control area was located 15 m from the farthest upwind edge of spray line 2. Collector locations in diagram are not representative of actual distances in the field.

Hobo temperature and RH data loggers were also used at different heights (2.5 and 9.2 m) to monitor atmospheric stability. Atmospheric stability was categorized by first calculating the stability ratio with the following equation from Fritz (*Fritz, 2006*):

$$SR = \left[ \frac{(TZ_2 - Tz_1)}{\mu w^2} \right] \times 10^5 \tag{1}$$

where SR is the stability ratio, $TZ_2$ and $Tz_1$ is the air temperature (°C) at 9.2 and 2.5 m, respectively, $\mu w$ is the mean wind velocity (cm/s) at 2.5 m, and $10^5$ gives the ratio acceptable units. The SR for each treatment application was calculated from the average wind speed and temperatures and were assigned to four categorical variables, "unstable," "neutral," "stable," and "very stable" (*Fritz, 2006*).

Treatments consisted of two formulation types, two adjuvants, and water only. Each formulation type and adjuvant was applied individually (four treatments), and each formulation type was also combined with each of the adjuvants (four treatments). This results in nine treatments, when including water only, from a 3×3 factorial design. We used the active ingredient spinosad (a mixture of spinosyn A and spinosyn D) formulated as a wettable powder (WP) and a suspension concentrate (SC) under the product names Entrust® insecticide and Entrust® SC insecticide containing 80% and 22.5% active ingredient, respectively (Dow AgroSciences, LLC, Indianapolis, IN, USA). The two adjuvant products were Maximizer (Loveland Products, Inc., Loveland, CO, USA) and Powerlock (Winfield Solutions, LLC, Arden Hills, MN, USA). Maximizer is composed of paraffin-based petroleum oil (83%) and nonionic surfactant (16.3%), whereas Powerlock contains modified vegetable oil (63%) and nonionic surfactant (32%). The water-soluble fluorescent dye Rhodamine-WT (CAS No: 37299-86-8) was mixed with the spray solution of each treatment (0.2% v/v) to allow for the quantification of spray deposition (*Smart & Laidlaw, 1977*).

Petri dishes and spinners were placed within the control area at the beginning of each replication and the treatment solution was sampled immediately before application to measure the actual fluorescent dye concentration in the tank. From the tank sample, two control dishes were loaded with 0.1 mL to quantify potential tracer degradation and recovery for each replication. Eight Petri dishes were also placed within the spray swath before application to estimate deposition within the targeted area. These samplers were evenly spaced 29.3 m apart along the middle of each of the two spray lines.

Treatments were selected in random order and applications were made with a ground rig boom sprayer (Avenger high clearance tractor; LeeAgra, Lubbock, TX). The boom length was 7.62 m and all applications were made with a boom height above the ground of 58 cm. For this study, 15 TeeJet XR11002 flat fan broadcast nozzles with a 110° spray angle, and a size 50 mesh (TeeJet 8079-PP-50), were evenly spaced 50.8 cm apart, along the length of the boom. Spray pressure was measured at the end of the spray boom and driving speed was monitored by an onboard GPS system; these were held at 2.07 bar and 6.8 km/h (4.2 mph), respectively. Two spray passes per application were made to simulate a single pass with a 15 m spray boom.

Following each treatment replication (spray application) horizontal and vertical samples were collected from all locations (off-target, on-target, and untreated). Sample collection began three minutes after the sprayer had been turned off at the end of the spray swath to allow for deposition to occur at the farthest samplers. Exposed Petri dishes and microscope slides were placed in dark containers to minimize photo-degradation of any insecticide or fluorescent material, and control samples were the last to be collected. Replications were performed over time within the same day or over multiple days with nine replications of each of the nine treatments, totaling 81 spray events in the months of July (2014), and May (2015, 2016).

### Field trial analysis

Insecticide deposition on Petri dishes was extracted with 15 mL of deionized (D.I.) water and decanted into 20-mL scintillation vials (Thermo Fisher Scientific, Waltham, MA, USA). Analysis vials were wiped with Kimwipes (Kimberly-Clark, LLC, Roswell, GA, USA) to

**Table 1 Sample areas for in-swath and downwind ground deposition samplers.**

| Swath | | | Downwind | | |
|---|---|---|---|---|---|
| Sample location (m) | Range (m) | Sample area (given a sampler diameter of 14 cm) (cm$^2$) | Sample location (m) | Range (m) | Sample area (given a sampler diameter of 14 cm) (cm$^2$) |
| 18.13 | 0–36.25 | 50,750 | 1 | 0–1.5 | 2,100 |
| 54.38 | 36.25–72.5 | 50,750 | 2 | 1.5–3 | 2,100 |
| 90.63 | 72.5–108.75 | 50,750 | 4 | 3–6 | 4,200 |
| 126.88 | 108.75–145 | 50,750 | 8 | 6–12 | 8,400 |
| | | | 16 | 12–24 | 16,800 |
| | | | 32 | 24–32 | 11,200 |

remove exterior moisture, and were inspected for clarity before being analyzed. Light absorption at a specific wavelength, representing the amount of dye present in each sample, was quantified and recorded using a GFL-1A fluorometer (Opti-Sciences, Inc, Hudson, NH, USA). The source and detection filters that were used for the excitation and detection of Rhodamine-WT were 530 and 590 nm, respectively (manufacturer recommendation). Standard curves were prepared using serial dilutions prepared in D.I. water. The detection limit (DL) of Rhodamine-WT, given the sensitivity of the fluorometer, was estimated by adding three standard deviations of a known low concentration measured 20 times to the mean of a blank sample (*Armbruster & Pry, 2008*). After extraction and analysis, Petri dishes were discarded and scintillation vials were triple rinsed with D.I. water before reuse. Less than 10% of the ground samples consisted of concentrations below the DL so one half of the DL was substituted for non-detectable concentrations (*Lubin et al., 2004*; *Helsel, 2005*; *Schleier et al., 2012*).

Statistical analysis for ground deposition was conducted using the amount of Rhodamine-WT deposited per unit area (μg/cm$^2$) for each distance, averaged over the two sample lines. Tank mixes for each treatment were analyzed for actual dye mixing rates following the procedure above. Based on the actual amount of dye in the tank, the estimated volume on the spiked control plates was compared to the actual volume of the spike (0.1 mL) to estimate recovery rates for each treatment. The following steps were followed to represent deposition as a percentage of the total material applied (*Fritz et al., 2011*). First the area for each sample location was calculated, as one-half the distance between two sample locations multiplied by the diameter of the Petri dish (Table 1). The application rate of the dye for each treatment was then estimated by multiplying the actual dye mixing rates by the application rate (Table 2). Multiplying the application rate of the dye by the total area of the sample locations returned the total mass of dye applied. The area for each sample location was then multiplied by the deposition data, averaged over the two sample lines, for each distance, to return mass of dye per sample location. Finally, the mass of dye per sample location was divided by the total dye applied to express deposition as a percentage of applied material. Deposition data for both in swath and downwind samples were calculated in this way.

**Table 2 Application rates of Rhodamine-WT (RWT) for each treatment[a].**

| Treatment | Dye mixing rate (µg/mL) | Flow rate (L/ha) | RWT application rate (µg/cm$^2$) |
|---|---|---|---|
| Entrust SC insecticide[a] | 80.06 | 123.47 | 0.0988 |
| Entrust insecticide[b] | 96.46 | 127.21 | 0.1227 |
| Entrust SC & PowerLock | 103.15 | 127.21 | 0.1312 |
| Entrust SC & Maximizer | 72.35 | 128.15 | 0.0927 |
| Entrust & PowerLock | 93.26 | 130.02 | 0.1213 |
| Entrust & Maximizer | 83.38 | 127.21 | 0.1061 |
| PowerLock[c] | 93.69 | 124.41 | 0.1165 |
| Maximizer[c] | 80.41 | 124.41 | 0.1000 |
| Water | 82.07 | 127.21 | 0.1044 |

Notes:
[a] Suspension concentrate formulation of the insecticide spinosad.
[b] Wettable powder formulation of the insecticide spinosad.
[c] Spray enhancement additives.

Due to minimal ground deposition beyond 32 m, only six downwind distances were included in the statistical analysis. This resulted in a total of 486 data points (nine treatments, six distances, nine reps) for ground deposition which were analyzed using multiple linear regression in the statistical software package R, version 3.3.2 (*R Development Core Team, 2016*). Exploratory data analysis was done to identify outliers, potential interactions, linearity, and normality, among dependent and independent variables. Table 3 lists summary statistics for numerical variables considered in this data set.

Log transformations on the deposition of Rhodamine-WT (µg/cm$^2$) and the independent variable, distance (m), were required before parametric statistics could be used. Correlation coefficients between independent variables was used to eliminate those with a correlation greater than 0.5 to avoid collinearity. This resulted in the exclusion of temperature and RH at the upper height (9.2 m), as well as temperature measured at the lower height (2.5 m). Temperature, instead of RH, at 2.5 m was removed because RH resulted in a better fit for the regression model, and because 2.5 m is a more practical measurement height. Model selection with Akaike's information criterion (AIC) was used to select between candidate models, which were based on specific hypotheses about pesticide drift. Full and reduced models were compared using an extra sums of squares (ESS) *F*-test to determine the contribution of certain independent variables in explaining variability in the response variable. The model with the lowest AIC was selected and diagnostic plots were used to check that the requirements for linear regression were met regarding statistical assumptions. The data were centered by subtracting the average RH so that the main term coefficients for treatment could be interpreted as the estimates at average RH, rather than zero.

Droplet spectra resulting from deposition on vertically positioned microscope slides were recorded with a DropVision measurement system (trademark of Leading Edge Associates, Inc.). The DropVision system integrates a compound microscope and image processing software to recognize, count, and measure droplets while eliminating background objects. A calibration slide containing circles of known diameters was used to

**Table 3 Summary statistics for numerical variables.**

| Variable | Units | Mean | SD | Range | |
|---|---|---|---|---|---|
| | | | | **Minimum** | **Maximum** |
| Rhodamine-WT | $\mu g/cm^2$ | $2.88 \times 10^{-3}$ | $6.27 \times 10^{-3}$ | $2.60 \times 10^{-7}$ | $5.08 \times 10^{-2}$ |
| Active ingredient | $\mu g/cm^2$ | 0.0236 | 0.0619 | $6.05 \times 10^{-7}$ | 0.6938 |
| Tank solution | $\mu L/plate$ | 5.025 | 0.0109 | $4.50 \times 10^{-4}$ | 88.52 |
| VMD | $\mu m$ | 36.7 | 5.08 | 26.08 | 52.54 |
| Wind speed | m/s | 2.27 | 0.6865 | 0.3 | 3.82 |
| Temp @2.5 m | °C | 20.39 | 4.14 | 13.38 | 30.14 |
| Temp @9.5 m | °C | 21.2 | 4.56 | 12.96 | 31.08 |
| RH @2.5 m | % | 58.55 | 12.23 | 26.94 | 82.08 |
| RH @9.5 m | % | 55.03 | 13.68 | 22.67 | 86.16 |
| Stability ratio | | 2.11 | 5.83 | −27.22 | 32.56 |
| Distance | m | | | 1 | 32 |

Note:
VMD, volume median diameter; Temp, temperature; RH, relative humidity.

calibrate the system at 10× magnification. This is achieved by relating the number of pixels contained in the calibration circle to its diameter. Microscope slides for all three years (4,221 slides) were scanned by a single person following a specific protocol to minimize user error. A specific viewing pattern was designed to sample a representative number of droplet impressions from an evenly distributed area of the slide surface. A minimum of 100 droplets or 200 pictures were required before moving to the second slide from a given field location (each spinner contains two slides). Once 200 droplets were counted, or the entire surface area of both slides was viewed, the droplet statistics were compiled into a report produced by the software. The diameter at which half of the volume was contained in droplets smaller than the median (VMD) was recorded for all slide sets containing 30 or more droplets, which limited the farthest downwind distance from the spray source to 32 m, resulting in 273 data points for statistical analysis.

The same statistical approach for ground deposition was used to analyze the droplet data. Correlated independent variables that were excluded above were also excluded here. Exploratory analysis suggested a log transformation of the response (VMD) was required to meet assumptions of normality and AIC model selection was used to choose the final model. The predictor variable for height was analyzed as a categorical variable at one and two m above the ground. Linear model assumptions were assessed using residual plots and the global validation package, gvlma, in R (*Slate, 2014*). These data were centered by subtracting the average wind speed so that the main term coefficients for treatment could be interpreted as estimates at average wind speed.

## Wind tunnel

The droplet spectra of all treatments were also measured in a wind tunnel so that general trends in droplet size could be compared to differences in ground deposition from the field study. Using the same application system as in the field study (i.e., nozzle set up, application rate, and spray pressure) three replications of each treatment were sprayed in a

**Table 4 Treatments ordered by droplet size and ground deposition[a].**

| Treatments containing active ingredient sprayed in wind tunnel | % Spray volume containing droplets ≤141 μm (SD) | Treatments ordered by ground deposition (highest to lowest) | Droplet size is indication of spray drift |
|---|---|---|---|
| WP | 19.98 (0.61) | WP | Yes |
| SC | 15.30 (0.28) | SC | Yes |
| SC & Maximizer | 13.62 (0.03) | SC & Maximizer | Yes |
| WP & Maximizer | 13.33 (0.08) | WP & Maximizer | Yes |
| WP & PowerLock | 12.18 (0.23) | WP & PowerLock | Yes |
| SC & PowerLock | 11.69 (0.08) | SC & PowerLock | Yes |

Notes:
WP, Entrust (wettable powder formulation of insecticide spinosad); SC, Entrust SC (suspension concentrate formulation of insecticide spinosad).
[a] The treatments with the largest fraction of "fine" droplets (100–175 μm) measured in the wind tunnel had the highest downwind ground deposition in the field study at any downwind distance.

wind tunnel at the University of Nebraska West Central Research & Extension Center in North Platte, Nebraska. A Sympatec laser diffraction particle size analyzer (Sympatec, Inc., Clausthal-Zellerfeld, Germany) positioned near the spray nozzle was used to measure droplet spectra for each of the treatments in the wind tunnel. Droplet sizing data measured for each treatment included VMD, the 10% and 90% diameters, the relative span, and the percent spray volume contained in droplets less than 141 μm. This percentage represents the fraction of spray droplets within the "fine" classification (100–175 μm) for droplet sizes (*ASABE, 2009b*), and was used as an indicator for spray drift. The treatments were ranked according to the percentage of droplets within this size range and compared to the ranks for ground deposition (Table 4).

## Efficacy

An insecticide efficacy experiment was also conducted to test differences in insect control between the treatments used in the drift study. The study was conducted August 2016 at the same experimental station in Fresno, California. The experimental setup was a randomized complete block design with seven treatments (treatments consisting of only adjuvant were excluded) and four replications (blocks) of each treatment. Each plot was 3.05 m long and 2.03 m wide, and contained two rows of newly planted broccoli (Green Magic variety of Brassicaceae sp., 16 plants per plot). Plots were 1.5 m apart, and two empty rows (2.03 m) were left between blocks to avoid contamination from adjacent plots. Before spraying on application day a pre-count was conducted to record cabbage looper larvae (*Trichoplusia ni*) on broccoli plants in the study area which had been infested by endemic populations. Applications were made with a handheld boom sprayer at the same rate, and with the same nozzle type, as in the drift study. All treatments were applied in random order and *T. ni* larvae on the broccoli plants were counted to estimate percentage mortality at 1, 3, and 7 days after application.

Efficacy data were represented and analyzed as count data, and as a fraction of the untreated control plots at each time-step after application. Count data and percent mortality data were analyzed using ANOVA fit to a linear model for a randomized block design in R, version 3.3.2 (*R Development Core Team, 2016*). The hypothesis that was tested with the

count data was that there would be no differences in the number of *T. ni* larvae between treated and untreated plots. For the percent morality data, the tested hypothesis was that there would be at least one difference in percent mortality between treatments.

## RESULTS

### Ground deposition

The recovery rates of Rhodamine-WT for each treatment were within the recommended range of 80–120% (*EPA, 2016a*; *ASABE, 2009a*). The high recovery rate of the dye from the control plates suggests that degradation of the fluorescent dye in samples was negligible due to the short exposure time to sunlight. In some cases, more material was deposited on the swath plates than was expected given the application rate of the dye. Potential sources of error that may have contributed to an inaccurate estimation of recovery include fluctuations in the actual driving speed, spray pressure, flow rate, or errors in the analysis on the fluorometer (*Arvidsson, Bergstrom & Kreuger, 2011*).

Coefficient estimates and standard errors for the selected model for ground deposition, centered on average RH, are listed in Table 5. Treatment, log of distance (m), wind speed (m/s), RH (%), and a term for the interaction between treatment and RH were included in the final model. Year, stability ratio, and stability category were excluded because they had no effect on deposition, and did not significantly change the error sums of squares when compared to the final model (ESS *F*-test, $F = 1.46$, $p = 0.2004$, on 461 and 466 degrees of freedom). Diagnostic plots of the model residuals suggested that the assumptions of normality, linearity, and homoscedasticity were reasonably met. The selected regression model for the ground deposition data was shown to explain 89.3% of the variability in the response variable (adjusted $R$-squared of 0.8934 for overall model). The regression equation centered on RH with water as the reference level for treatment is:

$$
\begin{aligned}
LT = {} & -5.43 + 0.305 * WP + 0.162 * SC - 0.660 * WPMax - 0.683 * WPPL \\
& - 0.491 * SCMax - 0.894 * SCPL - 0.543 * Max - 1.058 * PL - 1.662 * LD \\
& + 0.456 * Wind - 0.019 * RH + 0.006(WP * RH) - 0.038(Max * RH) \\
& + 0.034(PL * RH) + 0.0176(SCMax * RH) + 0.003(SCPL * RH) \\
& - 0.033(SC * RH) + 0.034(WPMax * RH) - 0.001(WPPL * RH)
\end{aligned}
\tag{2}
$$

where LT is the log of the Rhodamine-WT deposition ($\mu g/cm^2$), WP = Entrust (WP) insecticide, Max = Maximizer, SCPL, Entrust SC insecticide with PowerLock, WPMax, Entrust (WP) with Maximimzer, WPPL, Entrust (WP) with PowerLock, LD is the log of the distance (m),Wind is wind speed (m/s), and RH is the RH (%) at 2.5 m above the ground. Full coefficients for Eq. (2) are listed in Table 5.

Deposition of Rhodamine-WT ($\mu g/cm^2$) for all treatments decreased with increasing distance from the field edge as indicated by the negative coefficient for logged distance in the overall regression model ($p < 0.0001$, Table 5). After controlling for wind speed and RH, this decrease in deposition was estimated to be 68.40% with an associated 95% confidence interval (CI) from 67.19% to 69.57% for every doubling of distance (e.g., going from one ground sample to the next in this study, Fig. 2). The treatments with the highest and lowest

**Table 5 Coefficient estimates and SEs for selected ground deposition model[a].**

| Ref. = Water | Estimate | Std. error | t-value | Pr(>\|t\|) |
|---|---|---|---|---|
| (Intercept) | −5.42587 | 0.20664 | −26.258 | <0.0001* |
| WP | 0.304542 | 0.146499 | 2.079 | 0.038* |
| SC | −0.16177 | 0.166646 | −0.971 | 0.332 |
| Maximizer[b] | −0.54323 | 0.150797 | −3.602 | 0.0003* |
| PowerLock[b] | −1.05842 | 0.148866 | −7.11 | <0.0001* |
| SC Maximizer | −0.49139 | 0.153832 | −3.194 | 0.001* |
| SC PowerLock | −0.89444 | 0.146654 | −6.099 | <0.0001* |
| WP Maximizer | −0.65979 | 0.160134 | −4.12 | <0.0001* |
| WP PowerLock | −0.68294 | 0.159084 | −4.293 | <0.0001* |
| Log distance (m) | −1.66215 | 0.02758 | −60.267 | <0.0001* |
| Wind speed (m/s) | 0.455592 | 0.063746 | 7.147 | <0.0001* |
| RH (%) | 0.019221 | 0.006816 | 2.82 | 0.005* |
| WP × RH (%) | −0.00602 | 0.01066 | −0.565 | 0.572 |
| SC × RH (%) | −0.03335 | 0.021728 | −1.535 | 0.125 |
| Maximizer × RH (%) | −0.03756 | 0.014681 | −2.558 | 0.010* |
| PowerLock × RH (%) | 0.034091 | 0.010877 | 3.134 | 0.001* |
| SC Maximizer × RH (%) | 0.017643 | 0.016531 | 1.067 | 0.286 |
| SC PowerLock × RH (%) | 0.002819 | 0.009565 | 0.295 | 0.768 |
| WP Maximizer × RH (%) | 0.03434 | 0.014663 | 2.342 | 0.019* |
| WP PowerLock × RH (%) | −0.00109 | 0.009062 | −0.121 | 0.904* |

Notes:

WP, Entrust (wettable powder formulation of insecticide spinosad); SC, Entrust SC (suspension concentrate formulation of insecticide spinosad); RH, relative humidity.

[a] Data centered on mean RH so that the estimates of the main term effects can be interpreted at average RH instead of zero. Overall model had an adjusted $R^2$ of 0.8934.

[b] Spray enhancement additives.

* Statistically significant at $\alpha = 0.05$.

amounts of active ingredient observed at the farthest downwind distance were Entrust SC insecticide with Maximizer and Entrust (WP) insecticide with PowerLock, respectively.

All treatments also exhibited more deposition in higher wind conditions as indicated by the negative coefficient for wind speed in the overall regression ($p < 0.0001$, Table 5), with every additional unit increase in wind speed (m/s) resulting in an estimated 36.59% increase in deposition after controlling for distance and RH (95% CI from 28.13% to 44.06%) (Fig. 3).

The interaction between RH and treatment suggests that the effect of RH on deposition depends on treatment type. Neither Entrust SC nor Entrust (WP) were affected by RH, but water alone, as well as all treatments which included adjuvant, exhibited higher deposition with increasing RH over the range of RH measured in this study.

Relative humidity had a larger positive effect on deposition of Entrust SC with Maximizer compared to the Entrust SC treatment alone ($p = 0.041$). However, the effect of RH on deposition of the Entrust SC with PowerLock combination was not significantly different than for Entrust SC alone ($p = 0.10$). The same trend was observed when comparing the effect of RH on deposition between the WP formulation and WP plus the

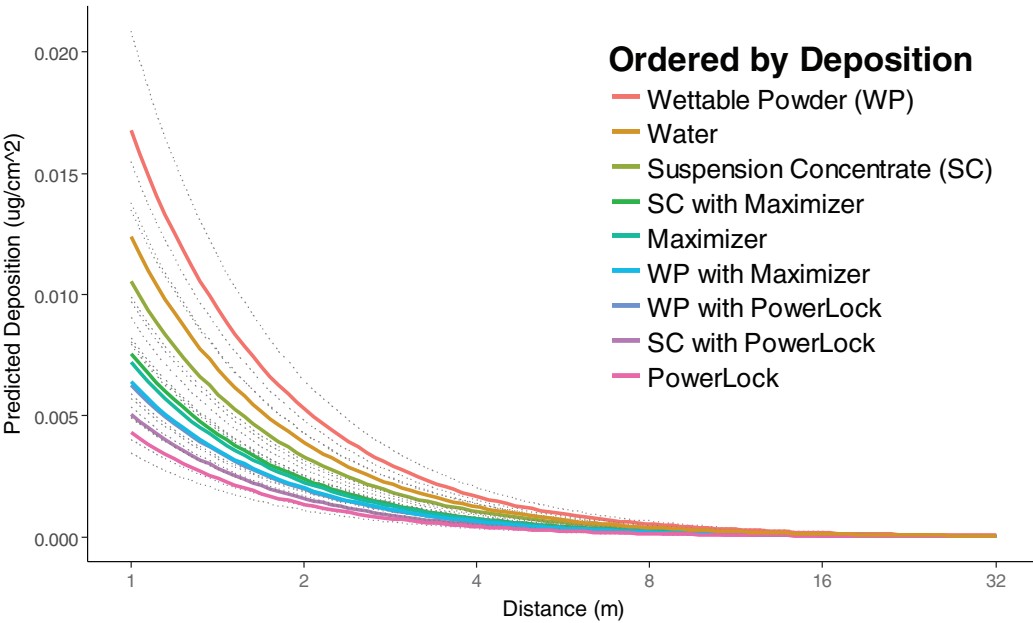

**Figure 2 Predicted deposition of Rhodamine-WT as a function of distance at average RH and wind speed.** After controlling for wind speed and RH, this decrease in deposition was estimated to be 68.4% with an associated 95% confidence interval from 67.19% to 69.57% for every doubling of distance (i.e., going from one ground sample to the next in this study). Adjusted $R^2$ of 0.8934 from overall model and dotted lines represent the 95% CI.

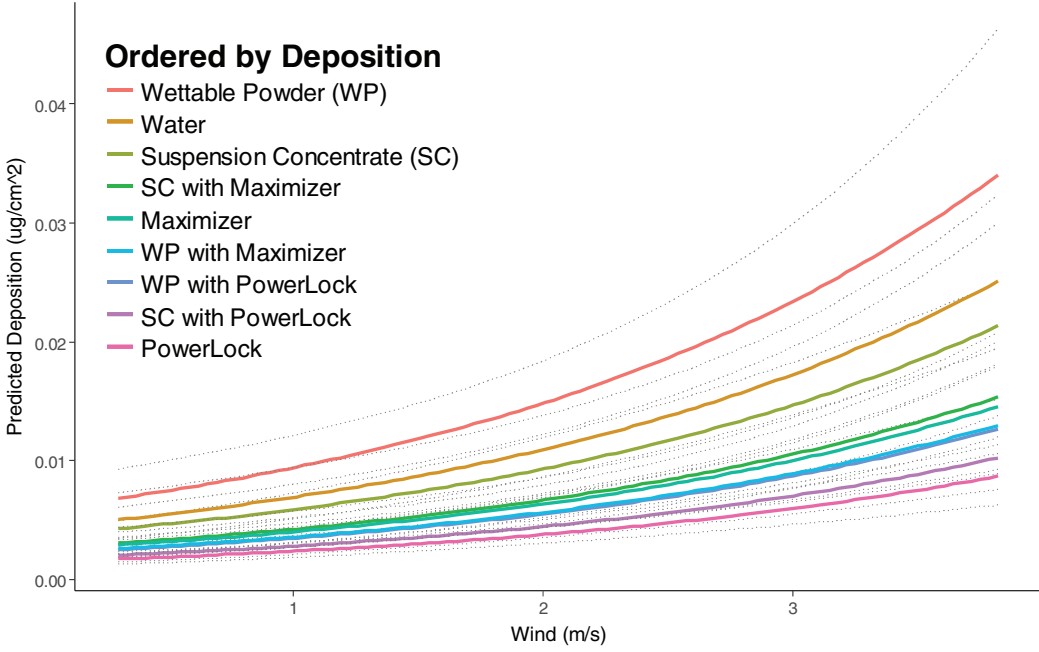

**Figure 3 Predicted deposition of Rhodamine-WT as a function of wind speed at average RH and a distance of one m.** All treatments exhibited more deposition in higher wind conditions with every additional 1-unit increase in wind speed (m/s) resulting in an estimated 36.59% increase in deposition after controlling for distance and RH (95% CI from 28.13% to 44.06%). Adjusted $R^2$ of 0.8934 from overall model and dotted lines represent the 95% CI.

two adjuvants. Deposition of Entrust WP with Maximizer was more affected by RH than Entrust WP alone ($p = 0.010$), but Entrust WP with PowerLock was not affected differently than Entrust WP alone ($p = 0.623$).

At the average RH (58.55%) recorded in this study, deposition of the SC formulation was lower than for the WP formulation after controlling for distance and wind speed ($p = 0.0045$). The estimated difference in deposition between Entrust SC and Entrust WP was 37.27% with an associated 95% CI from 13.54% to 54.48%. With the exception of the Entrust WP and Entrust SC insecticide treatments, deposition of the water only treatment was greater than for all other treatments at the average RH, and fixed distance and wind speed ($\alpha = 0.05$).

The difference in deposition between the formulated product with and without the adjuvant is interpreted as the effect on drift caused by the adjuvant. When comparing deposition of the Entrust SC treatment to deposition of this formulation with each of the two adjuvants, PowerLock reduced deposition, but Maximizer did not ($p < 0.001$ and $p = 0.0594$, respectively). The estimated reduction in deposition caused by the addition of PowerLock to the SC formulation was 51.94% at average RH, and after controlling for distance and wind (95% CI from 33.33% to 65.35%).

Deposition of the WP formulation was reduced with the addition of both the PowerLock and the Maximizer adjuvants by roughly the same amount, at average RH ($p < 0.0001$). After controlling for distance and wind speed, this reduction was an estimated 62.75% (95% CI from 50.34% to 72.06%) and 61.88% (95% CI from 48.46% to 71.80%) for PowerLock and Maximizer, respectively.

## Vertical deposition

Coefficient estimates and standard errors for the selected model for VMD, centered on average wind speed, are listed in Table 6. Treatment, log of VMD (µm), wind speed (m/s), and RH (%), were included in the final model for vertical deposition. Year, stability ratio, and stability category were excluded because they had no effect on deposition, and did not significantly change the error sums of squares when compared to the final model (ESS $F$-test, $F = 1.204$, $p = 0.3079$, on 247 and 252 d.f.). Diagnostic plots of the model residuals suggested that the assumptions of normality, linearity, and homoscedasticity were reasonably met. The selected regression model for the VMD data was shown to explain 41.9% of the variability in the response variable (Adjusted $R$-squared of 0.4187 for overall model). The estimated regression equation for logVMD from the selected model centered on wind speed and with water as the reference level for treatment is:

$$
\begin{aligned}
\mathrm{lVMD} = {} & 3.941 + 0.133 * \mathrm{WP} + 0.119 * \mathrm{SC} + 0.109 * \mathrm{WPMax} + 0.093 * \mathrm{WPPL} \\
& + 0.089 * \mathrm{SCMax} + 0.149 * \mathrm{SCPL} + 0.073 * \mathrm{Max} + 0.135 * \mathrm{PL} - 6.9E \\
& - 4 * D - 0.005 * \mathrm{RH} - 0.070 * \mathrm{Height} + 0.101 * \mathrm{Wind} + 0.125(\mathrm{WP} * \mathrm{Wind}) \\
& + 0.094(\mathrm{SC} * \mathrm{Wind}) + 0.018(\mathrm{WPMax} * \mathrm{Wind}) - 0.030(\mathrm{WPPL} * \mathrm{Wind}) \\
& + 0.081(\mathrm{SCMax} * \mathrm{Wind}) + 0.085(\mathrm{SCPL} * \mathrm{Wind}) - 0.044(\mathrm{Max} * \mathrm{Wind}) \\
& + 0.072(\mathrm{PL} * \mathrm{Wind})
\end{aligned}
\tag{3}
$$

**Table 6 Coefficient estimates and SEs for selected volume median diameter (VMD)[a].**

| Ref. = Water | Estimate | Std. error | t-value | Pr(>\|t\|) |
|---|---|---|---|---|
| (Intercept) | 3.94112 | 0.049635 | 79.402 | <0.0001* |
| WP | 0.13277 | 0.03384 | 3.923 | 0.0001* |
| SC | 0.118991 | 0.029544 | 4.028 | <0.0001* |
| Maximizer | 0.073092 | 0.031878 | 2.293 | 0.023* |
| PowerLock | 0.134597 | 0.031834 | 4.228 | <0.0001* |
| SC Maximizer | 0.088095 | 0.030629 | 2.876 | 0.004* |
| SC PowerLock | 0.149319 | 0.030304 | 4.927 | <0.0001* |
| WP Maximizer | 0.108916 | 0.029193 | 3.731 | 0.0002* |
| WP PowerLock | 0.093255 | 0.037198 | 2.507 | 0.013* |
| Distance (m) | −0.00069 | 0.00068 | −1.02 | 0.309 |
| RH (%) | −0.00546 | 0.000679 | −8.042 | <0.0001* |
| Height (m) | −0.07025 | 0.013066 | −5.377 | <0.0001* |
| Wind speed (m/s) | 0.007064 | 0.035475 | 0.199 | 0.842 |
| WP* Wind speed (m/s) | 0.124947 | 0.061732 | 2.024 | 0.044* |
| SC* Wind speed (m/s) | 0.093626 | 0.041187 | 2.273 | 0.024* |
| Maximizer × Wind speed (m/s) | −0.04367 | 0.053981 | −0.809 | 0.419 |
| PowerLock × Wind speed (m/s) | 0.071954 | 0.050305 | 1.43 | 0.154 |
| SC Maximizer × Wind speed (m/s) | 0.081286 | 0.04373 | 1.859 | 0.064 |
| SC PowerLock × Wind speed (m/s) | 0.084699 | 0.058615 | 1.445 | 0.149 |
| WP Maximizer × Wind speed (m/s) | 0.01789 | 0.042898 | 0.417 | 0.677 |
| WP PowerLock × Wind speed (m/s) | −0.03036 | 0.050863 | −0.597 | 0.551 |

Notes:

WP, Entrust (wettable powder formulation of insecticide spinosad); SC, Entrust SC (suspension concentrate formulation of insecticide spinosad); RH, relative humidity.

[a] Data centered on mean wind speed so that the estimates of the main term effects can be interpreted at average wind speed instead of zero. Overall model had an adjusted $R^2$ of 0.4187.

* Statistically significant at $\alpha = 0.05$.

Where lVMD is the log of the VMD (μm), $D$ is the distance from the field edge (m), and height is the vertical distance above the ground from 1 to 2 m, at which the rotating microscope slides were positioned. All other variables are defined the same as in Eq. (2). Full coefficients for Eq. (3) are listed in Table 6.

The VMD was not significantly affected by distance after controlling for height, wind speed, and RH ($p = 0.3087$). Elevated RH led to smaller VMD values for all treatments at fixed height and wind speed as indicated by a negative coefficient from the overall regression model ($p < 0.0001$). The estimated rate at which VMD decreased was 0.544% for every 1-unit increase in RH (%) with an associated 95% CI from 0.411% to 0.677%. The results also suggest that larger droplets were collected on spinners at the lower height (one m) compared with the upper heights (two m) for all treatments and distances ($p < 0.0001$). Droplets collected at 1m were an estimated 6.7% larger than droplets collected at two m with an associated 95% CI from 4.35% to 9.15%.

The interaction between treatment and wind speed suggests that the effect of wind speed on droplet size differs between treatments. Of the treatments that were affected by wind speed, the effect was such that higher wind speed resulted in larger VMD values

(positive slope). The effect of wind speed on VMD was not different between Entrust SC insecticide and Entrust (WP) insecticide ($p = 0.568$). When the SC formulation was combined with either of the adjuvants the degree to which wind affected VMD was not changed ($p = 0.861$ and $p = 0.714$ for Entrust SC with PowerLock and Entrust SC with Maximizer, respectively). When the two adjuvants were added to the WP formulation the addition of PowerLock resulted in a lesser influence of wind speed on VMD ($p = 0.011$), but the effect on VMD was not different between Entrust WP with Maximizer and Entrust (WP) alone ($p = 0.058$). The effect of wind speed on VMD for water was significantly lower than for either formulations alone ($p = 0.024$ and $p = 0.044$, for Entrust SC and Entrust WP, respectively).

At the average wind speed (2.38 m/s), and after controlling for RH, distance, and height, there was no difference in VMD between the two formulations ($p = 0.666$), or between either formulation in combination with either of the adjuvants ($p = 0.283$; 0.272; 0.606; 0.175 for PowerLock and Maximizer combined with Entrust SC and Entrust WP, respectively). The VMD of water was significantly lower than all other treatment combinations at average wind speed and fixed values for RH, height, and distance. The VMD of water was an estimated 8.43% smaller than the treatment with the next largest VMD (Entrust SC with Maximizer), with an associated 95% CI from 2.74% to 13.79%. At the average wind speed there was no difference in VMD between either formulation with PowerLock or either formulation with Maximizer ($p = 0.110$ and $0.451$, respectively).

The fraction of spray volume containing droplets less than 141 μm, measured in the wind tunnel, can be viewed in Table 4. The order in which treatments had the highest to lowest fraction of fine droplets (<141 μm), measured in a wind tunnel, was the same for treatments ordered by decreasing ground deposition.

### Biological efficacy experiment

There were statistically fewer *T. ni* larvae in the treated plots than in the untreated plots at 3 and 7 days after application, suggesting that all treatments were effective at reducing the pest population ($p < 0.05$, ANOVA on 18 d.f.). Furthermore, no difference in percentage mortality was observed between treatments at either 3 or 7 days after application, suggesting that all treatments were similarly effective ($p > 0.05$, ANOVA on 15 d.f.).

## DISCUSSION

Our results provide information on deposition and environmental factors related to agricultural spray drift of two of the most commonly used formulation types (*Knowles, 2008*) under realistic application scenarios, including the use of enhancement additives. Overall, ground deposition values were within the range of EPA assumptions for drift (1–5%) for estimating pesticide exposure to adjacent areas when models are not used (*Felsot et al., 2011*). The fraction of applied material, and the estimated 68% decrease in ground deposition for every doubling of distance from the field edge, are comparable to findings from other drift experiments (*Asman, Jorgensen & Jensen, 2003*). Quantification of exposure and risk to non-target organisms using actual environmental concentrations from this, and similar studies, is needed to fully characterize the benefits of DRTs.

Our findings support previous studies in that formulation type can affect spray drift, and should be considered when evaluating a given spray system for its drift potential. Specifically, our results differentiate between two formulations that are typically categorized together with regard to their drift potential. Both WPs and SCs are formulations of solid crystalline active ingredients which form non-deformable dispersions throughout the spray solution (*Knowles, 2008*). These formulations likely share a common mechanism for affecting spray atomization on the basis of this physical property (*Hilz & Vermeer, 2013*). However, current scientific literature is inconclusive regarding the effect of solid dispersions on droplet size, and therefore drift (*De Schampheleire et al., 2009*; *Stainier et al., 2006*; *Hilz & Vermeer, 2012*; *Qin et al., 2010*; *Dexter, 2001*). This study provides evidence that spray solutions of formulations with solid particles do influence drift, and that drift of the WP formulation was greater than for sprays of water alone. Differentiating the drift potential of these closely related formulations could help inform DRT manufacturing decisions, although generalizations are premature.

The greater drift reduction of PowerLock compared to Maximizer in this study is consistent with results from *Western et al. (1999)*, who found that adjuvants of vegetable oil, rather than mineral oils, were more effective at reducing drift, but others have found the effect of these adjuvant types on VMD to be small (*Ellis, Tuck & Miller, 1997*). Given the many interactions between certain properties of the spray solution and other components of the system, it could be advantageous if no additional tank additives were required to improve drift reduction. Both of the adjuvants tested in this study were shown to effectively reduce deposition without any apparent tradeoffs with biological efficacy, but their effect depended on the formulation type with which they were combined. This demonstrates the additional level of uncertainty introduced by incorporating adjuvants marketed for drift reduction into spray solutions.

Deposition on the vertical samplers was used to characterize the size distribution of spray droplets throughout the off-target area. Over the distances that were sampled in this study, there was more of a vertical, rather than horizontal, stratification of droplet sizes, with larger droplets collected below two m heights. The discrepancy between droplet size and ground deposition with increasing distance could be explained by the fact that the total number of droplets at each distance was not quantified. It is likely that at the greater distances fewer droplets were contributing to both ground deposition and droplet size on the spinners. At average RH and wind speed, the treatment containing only water and Rhodamine-WT had the second highest ground deposition, and the smallest VMD. This suggests that smaller droplets resulted in greater drift, but this cannot be conclusively stated. To test the well supported assumption that smaller droplet sizes lead to greater drift (*Felsot et al., 2011*; *Miller, Butler Ellis & Lane, 2011*; *Al Heidary et al., 2014*; *Arvidsson, Bergstrom & Kreuger, 2011*; *Threadgill & Smith, 1975*; *Nuyttens et al., 2010*; *Carlsen, Spliid & Svensmark, 2006*), we analyzed the droplet spectra of our treatments in a wind tunnel. As expected, we found that treatments with smaller droplets correlated with greater off-target deposition in the field.

Meteorological factors that affected deposition were wind speed and RH. The observed effect of wind speed on drift is consistent with previous studies (*Felsot et al., 2011*;

*Fritz, 2006*; *Maybank, Yoshida & Grover, 1978*; *Smith, Harris & Goering, 1982*; *FOCUS, 2007*), and is further supported by the presence of larger droplets on the vertical samplers during higher wind conditions. The positive correlation between RH and deposition for some treatments in this study is also reasonable given the relatively short sampling distance from the field edge. Conditions with higher RH are less conducive to evaporation of spray droplets, which may have led to larger droplets and greater deposition over the distances sampled. This may still be true even though the relationship between RH and VMD observed in this study would suggest otherwise. The effect of RH was small relative to the effect of wind speed on VMD, with <0.1% decrease in droplet size for every 10% increase in RH, whereas a 10% increase in wind speed led to an estimated 1.3% increase in VMD.

## CONCLUSION

This research demonstrates that droplet size is an effective indicator of agricultural spray drift resulting from different formulation types and adjuvants. The EPA verification protocol currently stipulates that when the combined effect of nozzle design and formulated product is evaluated, the drift reduction rating is only valid for that specific combination (*EPA, 2016a*). Our results suggest that droplet size data could be used to demonstrate drift reduction regardless of the formulated product being sprayed, but more spray mixtures need to be tested before reference sprays can be defined for comparing and rating spray mixtures as DRTs.

®™ Trademark of The Dow Chemical Company ("Dow") or an affiliated company of Dow.

## ACKNOWLEDGEMENTS

Our sincere thanks to the staff at the DAS Western U.S. Research Center where the experimental field work took place. That work would not have been possible without the help of Dr. Byron Sleugh, Dr. Joe Armstrong, Dr. Claudia Kuniyoshi, Dr. Cristiane Müller, Noemi Thomson, Mark Muzio, Garrick Stuhr, and all the summer interns. A special thanks to Dr. Scott Hutchins and Dr. Monica Sorribas for advocating for this research at Dow (Corteva™) AgroSciences, from which financial support was generously provided.

### Funding

This research was funded by Dow AgroSciences, Contract No. 4W4789, "Assessing Efficacy, Exposure, and Risk for Pesticide Drift Reduction Technologies," the Montana Agricultural Experiment Station, and Montana State University. The funders had no role in study design, data collection and analysis, decision to publish, or preparation of the manuscript.

### Grant Disclosures

The following grant information was disclosed by the authors:
Dow AgroSciences: 4W4789.

Assessing Efficacy, Exposure, and Risk for Pesticide Drift Reduction Technologies. Montana Agricultural Experiment Station, and Montana State University.

## Competing Interests

C.J. Preftakes was an employee/student at Montana State University during the conduct of the study and is currently an employee of Bayer, Crop Sciences. J. Schleier was an employee of Dow AgroSciences, which is the primary registrant of spinosad, during the conduct of the study.

## Author Contributions

- Collin J. Preftakes conceived and designed the experiments, performed the experiments, analyzed the data, prepared figures and/or tables, authored or reviewed drafts of the paper, approved the final draft.
- Jerome J. Schleier III authored or reviewed drafts of the paper, approved the final draft.
- Greg R. Kruger contributed reagents/materials/analysis tools, authored or reviewed drafts of the paper, approved the final draft.
- David K. Weaver authored or reviewed drafts of the paper, approved the final draft.
- Robert K.D. Peterson authored or reviewed drafts of the paper, approved the final draft.

## Data Availability

The raw data for ground deposition and VMD are available in the Supplemental Files.

## Supplemental Information

Supplemental information for this article can be found online at http://dx.doi.org/10.7717/peerj.7136#supplemental-information.

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
