# Peer review of "Effect of insecticide formulation and adjuvant combination on agricultural spray drift"

_PeerJ, doi:10.7717/peerj.7136_

## Round 0.1 · original submission · Minor Revisions

· Academic Editor

Minor Revisions

The reviewers were complimentary of your work, but there are some comments that they would like to see you address in your revisions. Please note also that one of the reviewers has provided an annotated manuscript.

·

Basic reporting

The paper “Effect of insecticide formulation and adjuvant combination on agricultural spray drift” focuses on the role of some adjuvants as contributing factor to reduce the drift of sprayed insecticides. The topic is relevant and interesting, and the study contributes to the knowledge of the drift phenomenon. Results are clearly reported and coherently discussed. The text is clear and well understandable. In general, the paper is relevant for its publication on PeerJ.
Some minor changes for “Material and methods” and “Discussion” chapters and for the Figures are suggested to the Authors.

Experimental design

Experimental design, mainly based on Standard protocols, is accurately described and the statistical analysis of data seems correct. However, I need to clarify some points about the experimental methods. I am adding some comments for the Authors.

Validity of the findings

In general, the results mainly confirm previous findings on this topic; however, Authors clearly discuss both the points of novelty and of weakness of their work, coherently with the data obtained in the experiments.

Additional comments

• To improve the clarity of the paper, I suggest dividing the “Material and Methods” chapter (that is very long) in sub-paragraphs (for example: 1- field trial, 2 – ground deposition; 3- vertical deposition up to n – statistical methods).
• Some abbreviations have to be explained when they appear for the first time in the text (i.e. CI for Confidence Interval).
• Please, check superscript and subscript numbers in formulae and equations (H2O, cm2, etc).
• I think that the “Discussion” part can be partly reorganized, starting from some general statements about the work to more specific comments. Please, consider moving the lines 456-464 to the beginning of the “Discussion” section.
• Figures 2 and 3 are not very readable: it was difficult for me to distinguish the single curves and the grey area of the confidence interval make the lines less clear. I suggest trying to redesign the curves, for example plotting the CI with a fine bar than with an area.
• In Figure 1, I don’t find the black rectangles mentioned in the caption.
• In the Abstract (line 26) you mentioned that “the drift was reduced by as much as 37% by formulation type alone and reached 63% by incorporating certain spray adjuvant”. Firstly, I didn’t find the data of 37% reduction along the text. Please, could you put this finding more evident in the text or in a table that includes a summary of the relative effects (in %) of the treatments on the drift reduction? Then, I think that this sentence is clearer by dividing the effects of adjuvants on the drift from the intrinsic value of drift of each formulation type (it is a given characteristic of the formulation itself).
• Line 472: is it correctly placed?
• Line 151: what is a size of 50 mesh for the nozzle?
• Lines 182-184: is the area of each sampling point calculated according to a Standard protocol? Why the sampling points does not fall in the center of the range shown in Table 1? (I am sorry for the last dumb question, probably it depends from the fact that I didn’t catch perfectly the sampling scheme).
• Lines 263-269: in the efficacy test, you used a parametric method (ANOVA) to analyze the data of insect mortality. To me, it appears very improbable that this kind of data meet the assumption of normality. Please, check the data distribution normality and, if necessary, consider a non-parametric method to analyze data.

Reviewer 2 ·

Basic reporting

This submission meets all basic reporting criteria. It is especially well written and clear. The article includes sufficient introduction, however, the introductory paragraphs summarize a lot of detailed information that are not common knowledge for most readers, so a few more citations would be useful here.

Experimental design

This is a good piece of original research. The research rational, hypotheses and experimental approach are clearly detailed.

The methods are described in especially good detail and are very unambiguous. But here are a few points of clarification that would be useful.

First, lines 99-100 mention subsamples were averaged. Just to be clear, subsamples were not analyzed as part of the model? And why not if subsampling was used as part of the design?

Beginning line 164, please clarify timing of fluorometer measurements post-sampling and/or correction of results for degradation that may have occurred prior to analysis.

Log transformation for analysis and then reporting results for the untransformed finding are always problematic in terms of interpretation, especially when continuous findings are reported. For instance, line 306 and figure 2 are summarizing predicted trends in deposition as a function of distance – both were log transformed for analysis, so do the trends still hold as presented when untransformed? Similarly, the regression equation is developed for log VMD and results are reported for change in VMD as a function of RH (line 370) and windspeed (Figure 3) – how was transformation and back transformation of analysis handled? Some clarification in the methods or results would be useful.

Validity of the findings

There are no issues with the findings. The data are robust, statistically sound (see possible exception above), and controlled.

Additional comments

Nice piece of work. A lot of good deal of effort went into these results. I addition to comments here, the annotated manuscript points out a few minor things.

Annotated reviews are not available for download in order to protect the identity of reviewers who chose to remain anonymous.

---

## Round 0.2 · accepted · Accept

· Academic Editor

Accept

Nice job on the revisions and your responses to reviewer comments.